# Dynamic Response Identification of a Triple-Single Bailey Bridge Based on Vehicle Traffic-Induced Vibration Analysis

Vasileios D. Papavasileiou [ID], Charis J. Gantes *[ID], Pavlos Thanopoulos [ID] and Xenofon A. Lignos

Institute of Steel Structures, School of Civil Engineering, National Technical University of Athens, 15780 Athens, Greece
* Correspondence: chgantes@central.ntua.gr

**Abstract:** Even though prefabricated steel Bailey bridges have been used for more than 80 years, limited studies of their structural features have been conducted, most of which do not consider their response in operational conditions. This study aimed at determining the modal parameters of a 30.48 m length Triple-Single (TS) Bailey bridge based on traffic-induced vibrations and comparing them with numerical results. Low-cost improvised accelerometers recorded and logged the actual response time histories, while a three-dimensional (3D) numerical model was developed to carry out the relevant dynamic analyses. The identification of modal parameters was based on the Operational Modal Analysis (OMA) process and the Frequency Domain Decomposition (FDD) method. Numerical analysis results are in accordance with the operational dynamic response of the Triple -Single Bailey bridge, confirming that the numerical model can effectively be used for extended dynamic analysis. In addition, the analysis of raw time histories through the OMA process indicates that the response is affected by the connections' condition, in particular, the eventual looseness of bolts and pins. At least five eigenfrequencies were estimated and matched with relevant mode shapes.

**Keywords:** steel bridges; prefabricated bridges; bailey bridges; traffic induced vibration; low-cost sensors; automated FDD





## 1. Introduction

Prefabricated steel Bailey bridges [1] were developed and used during World War II (WWII) for temporary military use [2–4] and as a replacement during rehabilitation of damaged bridges. Nevertheless, many Bailey bridges are still operating today for civilian purposes, decades after their installation. At the same time, they are used to deal with emergencies for restoring bridges damaged by natural disasters. According to their use, prefabricated steel bridges [5] are classified as temporary, emergency, and permanent bridges, the latter having a life cycle design of 75 years. Except for permanent bridges, all other categories have a temporary duration and restrictions on their use. In the first years of use of such bridges, the restrictions on traveling speed and operational loads imposed by the competent supervisory authorities limited any structural issues, which were therefore of no concern to the engineers. However, the appearance of fatigue phenomena has led structural engineers to conduct research on prefabricated bridges since the 1960s.

Whitman and Alder (1960) [6] carried out the first experimental study of Bailey bridges, on which the British Department of Transport (1968) [7] relied for issuing technical guidelines for fatigue checking control. Webber (1970) [8] followed with an extensive experimental study on the fatigue strength of the bridge's panel. A fatigue strength study of panels was also carried out by Marsch and Barker (1988) [9]. An evaluation of the strength of a Double-Double Bailey 55 m bridge was carried out in 1990 [10]. Cullimore and Webber (2000) [11] reported new fatigue research on the Heavy Girder Bridge, an evolution of the Bailey bridge. King and Duan (2003) conducted experimental tests on scale models [12] for the maximum bearing capacity identification of the bridge. Parivallal

et al. (2005) [13] measured static semi-static loading of a Bailey bridge with strain and dial gauge instrumentation. King, Wu and Duan [14] (2013) studied a girder composed of two Bailey-type bridge panels to determine the ultimate loading capacity.

Khounsida, Nishikawa, Nakamura, Okumatsu and Thepvongsa [15] studied the static and dynamic response of three operating Bailey bridges and found discrepancies between raw field measurements and numerical results, with the estimation that they were due to member connections. One issue in the study was the limited available information on the bridges' cross-sections and the material's modulus of elasticity. Prokop, Odrobiňák, Farbák and Novotný [16] carried out numerical analyses on the loading capacity of Bailey bridges to determine the possibility of using them in civilian applications.

In most of the above studies, either sections of bridge girders or scaled models were used, while in those where numerical analyses were carried out, they were either static or quasi-static. No experimental research or in situ measurements have been carried out on Bailey bridges in operational conditions.

However, in the cases of dynamic tests, we are faced with two issues. The first issue refers to each vehicle's distinguishing spectrum "fingerprint", which is recorded during the tests and affects the analysis results. The second issue relates to the recorded time-history "noise" due to the deck roughness.

Recently a methodology to deal with these issues has been proposed [17–20], where the novel concept of developing a frequency-free vehicle to conduct dynamic tests helps to minimize the time history noise. An additional novelty of the proposed methodology is the use of one or a few sensors on the vehicle to record the time histories [19].

The main aim of this work was to record the dynamic response of an actual Bailey bridge operating for civilian purposes, hence subjected to uncontrolled passing of vehicles, under a "noisy" environment, through in situ measurements, and to develop and validate a finite element (FE) model with model updating [21] for use in future analyses.

## 2. Materials and Methods

### 2.1. Analysis Methods

A well-established method for determining the modal parameters of a structure (natural frequency, damping ratio, mode shape), examining also its dynamic response, is Experimental Modal Analysis (EMA) [22]. EMA is based on the analysis of imposed loads compared to the dynamic response of the structure. However, this practice is challenging in its application for large constructions, as heavy and expensive equipment is required [22,23].

Due to the disadvantages of EMA, the civil engineering community recently focused on a new experimental method for determining modal parameters, known as Operational Modal Analysis (OMA). According to OMA, to determine the modal parameters of the structure, it is necessary to record only the vibration response, which can be excited either from ambient forces or the operational loads, such as vehicle and pedestrian traffic [22]. The advantages are many, as the dynamic response of the bridge can be recorded freely without using special equipment excitation, while bridge traffic does not need to be interrupted.

There are two categories of OMA methods, Frequency Domain Methods and Time Domain Methods: both present advantages and disadvantages in terms of reliability of their results. According to Castellanos–Toros et al. [24], the advantage of analysis in the frequency domain is the simplicity and ease of the pertinent computational procedures. The main disadvantages in cases of complex structures refer to appearance of close modes. These methods are out of scope of the present study, and the reader can be informed extensively from the literature [22,25,26]. In the present study, the Frequency Domain Decomposition Method was applied, using the freely available Automated Frequency Domain Decomposition (AFDD) algorithm developed in [27,28].

### 2.2. Measuring Devices

To record accelerations under operational conditions with conventional methods, special laboratory equipment is required, which must be supported by experienced staff

and incurs high maintenance costs [21]. In this work, low-cost built-up sensors are used, controlled by only one operator, without affecting the traffic. The sensors measure the 3-axis accelerations, as it has been proven that accelerometer data are more useful for measuring vehicle–bridge dynamic interaction, leading to better estimates of vehicle impact [29].

Several researchers [24,30–33] have used improvised sensors for modal parameter identification with acceptable results. The six low-cost accelerometers developed for the needs of the current study are of the Micro Electromechanical Systems (MEMS) type. These sensors are designed to be entirely self-contained and free of wiring. Each sensor consists of an Arduino Uno (manufacturer: Arduino, Italy) microcontroller and an Inertial Measurement Unit (IMU) MPU 6050 (manufacturer: InvenSense Inc, USA) [34] and is powered by six AA 1.5 Volt batteries, while automatically logging data on a 16 GB Secure Digital memory card (SD Card). The schematic diagram is shown in Figure 1.

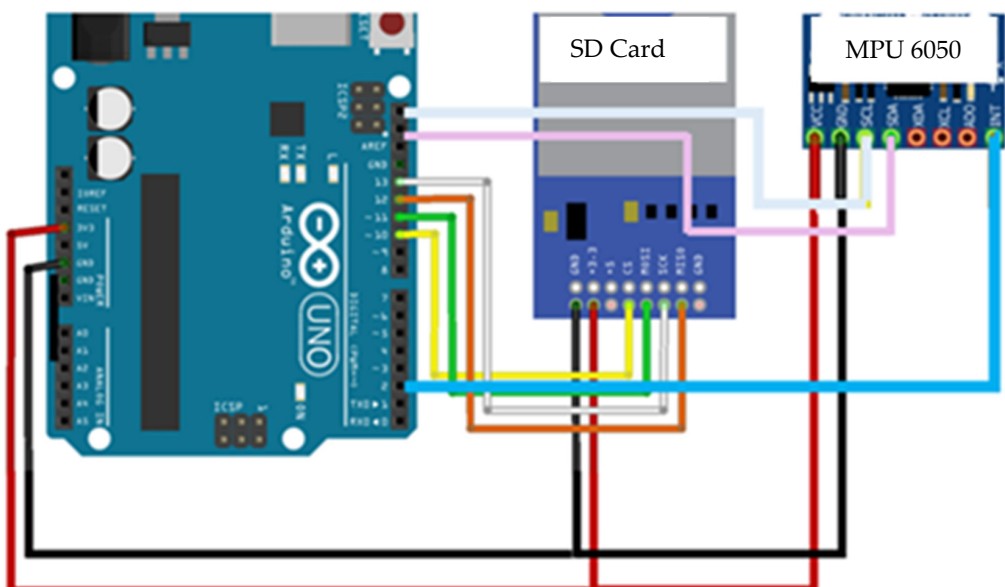

**Figure 1.** Accelerometer wiring schematic diagram.

Using open-source Arduino code, the frequency sampling (fs) is 500 Hz. Sensors of this type are inherently error prone, beyond the mentioned provisions to reduce the factors that cause noisy signals. While they are relatively reliable [35] for long-term recordings, the accumulation of these minor errors can grow without limitation, especially for low-cost MEMS [36].

The Allan Variance method is one of the most effective and straightforward methods developed to characterize noise in MEMS. According to David Allan [37], the dominant inertial errors and noise characteristics comprise constant bias, bias instability, velocity random walk for accelerometers, quantization, rate angle walk, rate ramp, and the sinusoidal component. According to Groves [38], regarding the classification of accelerometers, the MPU-6050 employed in this work belong to the category of tactical grade sensors, making them effective in measuring accelerations. The analysis of the present paper for MPU-6050 agrees with the results of other studies [39–42].

*2.3. Bailey Bridge Description*

The basic concept of the Bailey bridge is the formation of main girders by connecting prefabricated parts called panels (Figure 2). Panels are connected with pins to adjacent panels lengthwise at the four corners. The girders are placed on each side of the centerline (Figure 3).

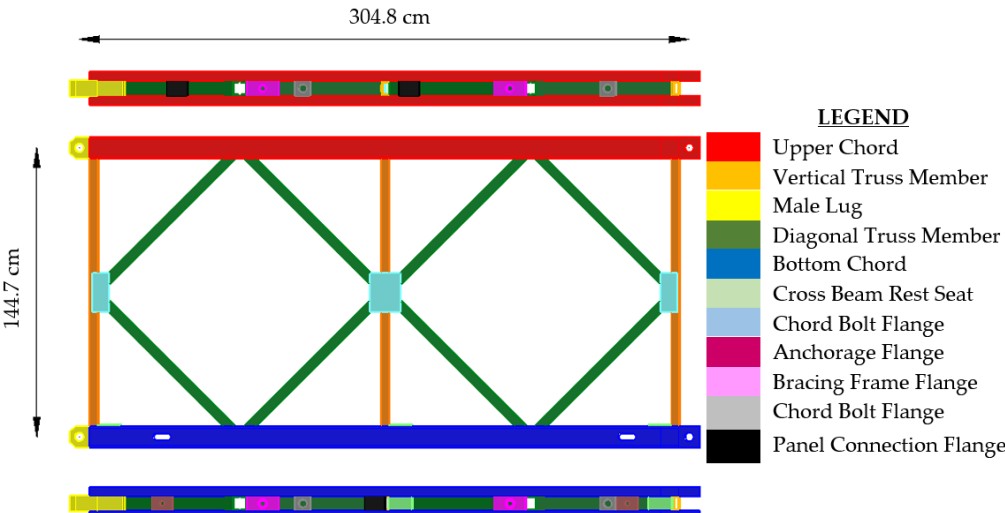

**Figure 2.** Typical M2 Bailey bridge panel (side view and plan views of chords).

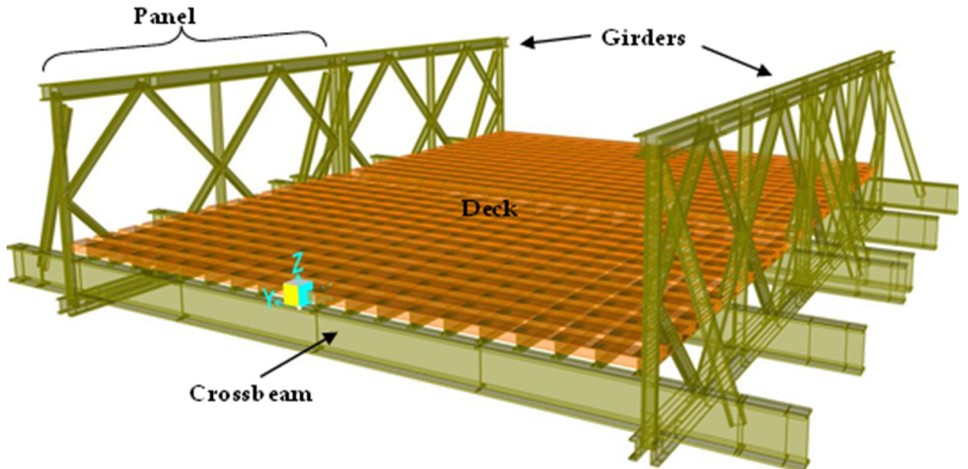

**Figure 3.** Bailey bridge segment ("bay" in US nomenclature).

Each panel is 304.8 cm long and 144.7 cm high. The cross beams connect the girders and support the stringers that are resting on their upper flange. The stringers, the wooden boards on top of them, and the curbs make up the deck. A segment of the bridge consisting of the panels, the partial girders, and the deck has a length of 304.8 cm. Many connections are required for the assembly of this type of bridge. Based on the original Bailey bridge, M1 in United State of America (USA) nomenclature, with a roadway up to 3.28 m, the USA Army developed the M2 alternative, having a wider roadway than M1, up to 3.81 m, and M3 with an even wider roadway up to 4.30 m [1,3]. The original design idea remains the same despite Bailey bridge's various transformations.

In Figure 4 the cross-sections of the seven primary Bailey bridge assemblies are shown, excluding the special ones that are not within the scope of this article. The first index, S, D, or T, refers to the number of side-by-side girders, while the second index refers to the number of girders in height. The arrows show the sequence in the initial assembly process and the required reinforcements, which increase the required loading capacity by adding girders in width and height.

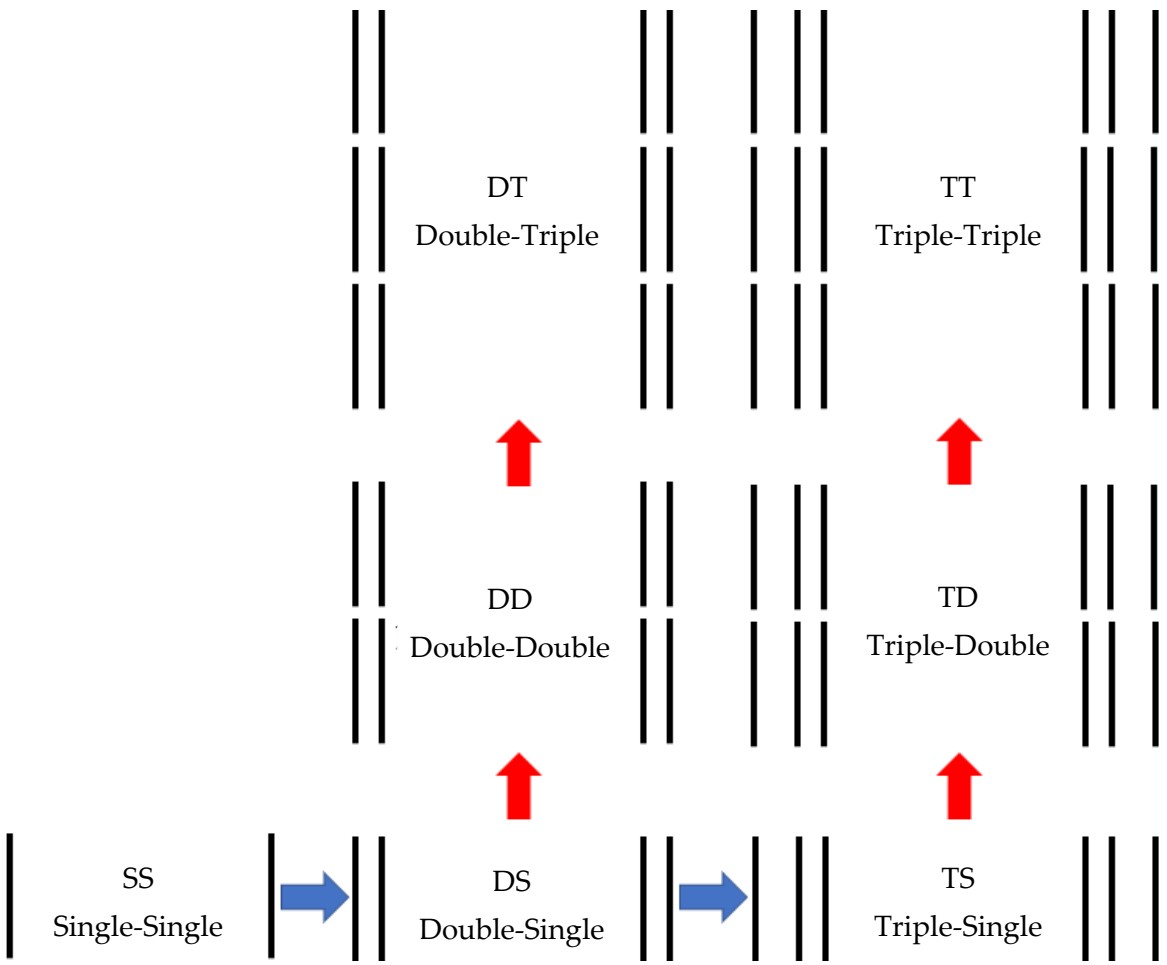

**Figure 4.** Bailey bridge assemblies, nomenclature, and abbreviations.

The nomenclature of the various Bailey bridge parts is assigned according to their origin (British or USA). Table 1 presents the differences in names according to their origin [1,3].

**Table 1.** Nomenclature of Bailey bridge parts.

| According to [1] (UK) | According to [3] (USA) | Usual Bridge Engineering Terminology |
|---|---|---|
| Stringers | Stringers | Stringers |
| Chord | Chord | Upper or lower horizontal member of panel |
| Cross girder | Transom | Cross beam |
| Raker | Raker | Vertical bracings |
| Sway brace | Sway brace | Horizontal bracings |
| Pin | Pin | Pin |
| Deck | Deck | Timber deck |
| Chord bolt | Chord bolt | Panel connection bolt [1] |
| Ribband | Ribband | Curb |
| Panel | Panel | Panel of truss girder |
| Gap | Gap | Span |
| Bay | Bay | Segment |
| End Post | End Post | End vertical member |

[1] for 2nd or 3rd story.

The present study focuses on the M2-type Bailey bridge, consisting of 29 parts. The assembled parts form simply supported bridges with lengths from 10 to 69 m. Two chan-

nels, welded back-to-back, form the panels' upper and lower cross-sections of horizontal members. The cross beams' cross-sections are formed from rolled steel joists (newly type cross-section belongs to IPE category). Table 2 presents the basic dimensions, features, and mass, and some of the cross-sections (vertical and diagonal panel members, cross beam), based on the on-site measurements and the literature references [43]. All table data refer to the M2 type bridge.

**Table 2.** Bailey bridge member sections.

| Member | Shape | Mass | Depth | Width | Thickness | | Area |
| --- | --- | --- | --- | --- | --- | --- | --- |
| | | | | | Web | Flange | |
| | | kg/m | cm | cm | cm | cm | cm$^2$ |
| Upper Chord | | 10.8 | 10.16 | 4.3 | 0.82 | 0.75 | 13.7 |
| Bottom Chord | Channel | 10.8 | 10.16 | 4.3 | 0.82 | 0.75 | 13.7 |
| Vertical/Diagonal | Channel | 6.1 | 7.62 | 3.50 | 0.43 | 0.69 | 7.81 |
| Cross Beam | Channel | 37 | 25.4 | 11.4 | 7.62 | 12.96 | 47.42 |
| Stringer [1] | I Section | 7 | 10.2 | 4.4 | 0.43 | 0.69 | 9.48 |
| Vertical Bracing | I Section | 6 | 4.06 | 6.35 | 0.41 | | 7.61 |
| Horizontal Bracing | I Section | 6 | | | Dia:2.9 cm | | 6.61 |

[1] Stringer is a panel composed of three I sections.

Three different steel grades are used in Bailey bridge parts [1,4]. The British Standard (BS) specifications for employed steel grades were known as BS 968, BS 15, and alloy steel (Manganese—Molybdenum) [43]. Indicatively, Table 3 lists the steel grades estimated to have been used in the bridge parts of the present study. It is crucial to be careful regarding steel properties for various analyses due to quality variations depending on the period of parts production. It is remarkable that Bailey [1] refers to steel properties with yield strength $f_y$ equal to 315 MPa and ultimate strength $f_u$ between 482 and 592 MPa, while the BS 968 determines strength limits (1962) [43], as shown in Table 3. In Table 4, the category of steel used in each bridge part is listed.

**Table 3.** Bailey bridge structural steel specifications.

| Quality | $f_y$ (MPa) | $f_u$ (MPa) | Thickness t (mm) | Remarks |
| --- | --- | --- | --- | --- |
| BS 968 [1] | 317.4 | $441.6 \leq f_u \leq 538.2$ | $t \leq 15.88$ | Replaced by BS 4360/1968 |
| BS 15 | 220.8 | $386.4 < f_u < 455.4$ | $t \leq 19.05$ | Replaced by BS 4360/1968 |
| Alloy steel | 897 | | | Manganese-molybdenum |

[1] Modulus of Elasticity E = 206.8 GPa.

**Table 4.** Steel property assignation on bridge parts.

| Parts Bridge Specified Nomenclature | BS 968 High Tensile Steel | BS 15 Mild Steel | Alloy Steel (Manganese-Molybdenum) |
| --- | --- | --- | --- |
| Panel (all members) | + | | |
| Cross Beam | + | | |
| Stringers | | + | |
| Horizontal Bracing | | + | |
| Bracing Frame Panel | | + | |
| Vertical Bracing | | + | |
| End Vertical | | + | |
| Pins | | | + |

The term Military Loading Class (MLC) refers to the loading capacity of the bridge, depending on its assembly (girder formation, span), determined according to [3,44–47]. The MLC calculation process is extensively described in the literature and differs between wheeled and tracked vehicles (battle tank vehicles). The MLC is equal to their weight

in short tons for tracked vehicles, while a calculation procedure is required for wheeled vehicles. The issue arises because the MLC does not include the civilian vehicles crossing the bridge. Therefore, in this case, and only for this study, the estimation based on the vehicle's weight (expedient method) is applied as described in the literature [45,46]. Table 5 shows the MLC in three traffic restriction cases.

**Table 5.** Bailey bridge cross restrictions.

| Traffic Restriction | Vehicle Position on Deck | Max Speed (km/h) | Min Spacing (m) | Authorization to Use |
|---|---|---|---|---|
| Normal | At any place | 40 | 27 | Anyone |
| Caution | On centerline | 13 | 45 | Supervised by the authorities |
| Risk | On centerline | 4 | One vehicle on bridge | Supervised by the authorities |

### 2.4. Bailey Bridge in Feneos (Corinth, Peloponnese, Greece)

The bridge evaluated in this study is a 30.48 m long Triple-Single Bailey bridge (Figure 5a,b and Table 6). It was built in 2019 to restore the regional network of the rural area when a previous concrete bridge was damaged following a natural disaster.

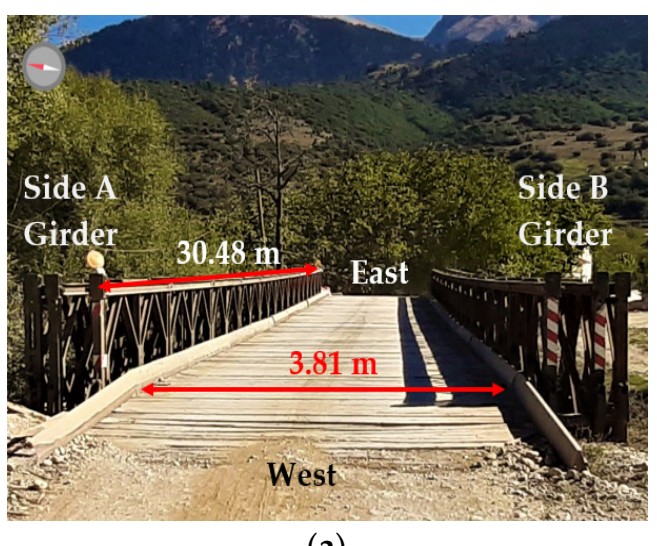

(a)

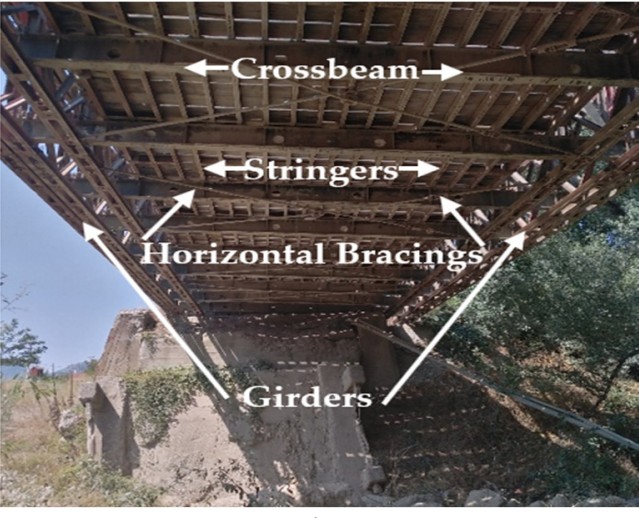

(b)

**Figure 5.** View (**a**) above; (**b**) below the bridge.

**Table 6.** TS bridge dimensions and weight.

| Bridge Type | Weight Per Fully Equipped Segment (kN) | Number of Fully Equipped Segments | Total Bridge Weight (kN) | Total Length (m) | Number of Effective lanes | Width of Effective Lane (m) |
|---|---|---|---|---|---|---|
| TS | 40.1 | 10 | 401 | 30.48 | 1 | 3.70 |

The bridge is supported on four independent seats, as shown in Figure 6. Each support consists of a pinned connection, which allows free rotation, and a stiffened plate that simply rests on the concrete floor, which allows horizontal movement if friction forces are overcome.

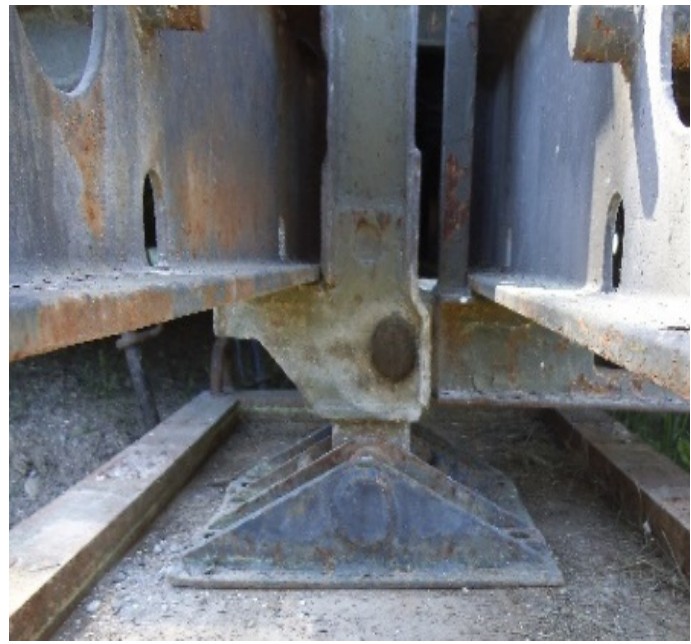

**Figure 6.** Bridge support.

Table 7 presents the bridge class, based on the literature [7], according to the crossing restrictions.

**Table 7.** TS-30.48 m Bailey Bridge Military Loading Class (MLC).

| Type of Vehicle | MLC Per Cross Restriction | | | Max Vehicle's Weight Based on Expedient Method (kN) | | | Remarks Expedient Method |
|---|---|---|---|---|---|---|---|
| | N [1] | C [2] | R [3] | N | C | R | MLC to kN = MLC × F1 × F2 × F3 |
| Wheeled | 50 | 57 | 60 | 558 | 635 | 670 | F1:1,15 MLC to short tons F2: 0.97 Short to metric tons |
| Tracked | 55 | 60 | 66 | 534 | 582 | 640 | F3: 10 metric tons to kN |

Normal [1], Caution [2], Risk [3]

*2.5. Bridge Instrumentation*

The sensors were arranged in such a way as to ensure the recording of the vertical and transverse components of the nodes' acceleration for the effective mode shapes estimation [48]. Three sensors were placed on each girder of the bridge. Distances and positioning information are presented in Table 8 and Figures 7 and 8.

**Table 8.** Sensor positioning information.

| Sensor | Segment | Panel | Position | Distance (m) |
|---|---|---|---|---|
| 1 | 1 | 1st side A (W-E) | Middle vertical member | 1.52 |
| 2 | 5 | 1st side A (W-E) | Third vertical member | 15.2 |
| 3 | 10 | 1st side A (W-E) | Middle vertical member | 28.9 |
| 4 | 1 | 1st side B (W-E) | Middle vertical member | 1.52 |
| 5 | 5 | 1st side B (W-E) | Third vertical member | 15.2 |
| 6 | 10 | 1st side B (W-E) | Middle vertical member | 28.9 |

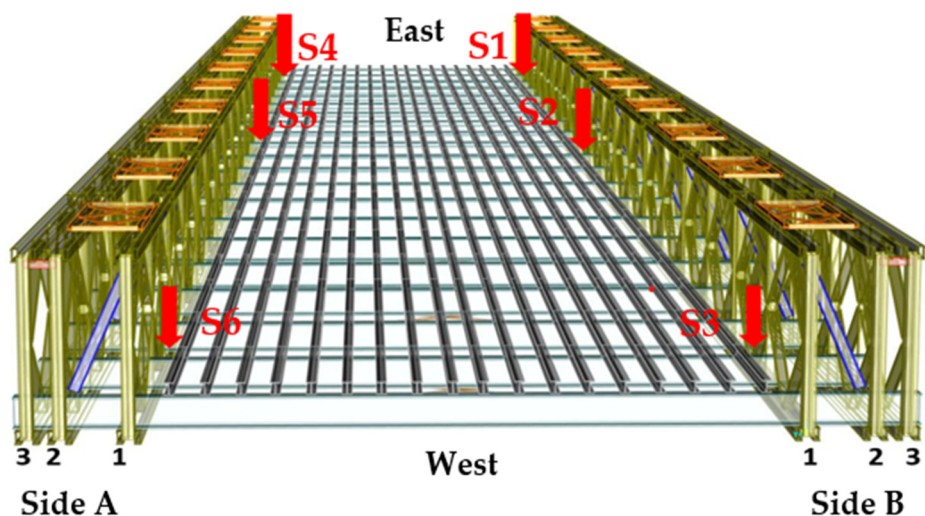

**Figure 7.** Positions of sensors.

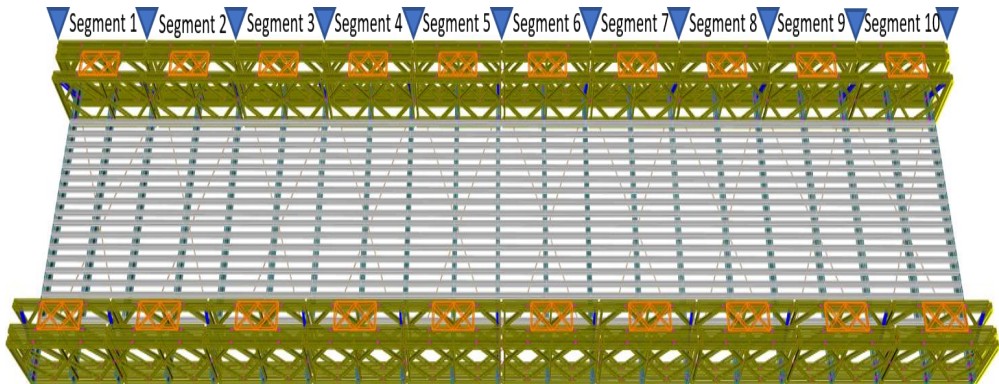

**Figure 8.** Bridge segments.

Each sensor can be used as a reference during the time history analysis process. So, sensors 1, 3, 4, and 6 were placed at nodes that do not coincide with nodes having zero displacements for the respective eigenforms, to record as many as possible [49]. The sensors were mounted to the bridge with four strong neodymium disc magnets (NdFeB) [48] of dimensions 2.0 cm (diameter) × 0.3 cm(thickness), with a holding force of 39 N each. Each sensor weighs up to 4.5 N.

### 3. Results

*3.1. Results of Time Histories Analysis Due to Traffic-Induced Vibration*

The sensors were used to record the response of the bridge over a period of 1735 s During this period, 27 heavy commercial and agricultural vehicles crossed the bridge. Information about the 27 vehicles is presented in Table 9. The term direction refers to the passing direction, west–east (W-E) and east–west (E-W).

Preprocessing was performed on the recorded time histories to ensure they were appropriate for analysis. Initially, all sensors' time histories were demeaned and zero padded. The processed total time histories of accelerations in the vertical direction are shown in Figure 9.

**Table 9.** Vehicle information.

| Index | Vehicle Type | Direction | Remarks |
|---|---|---|---|
| Veh1 | Farm truck | W-E | |
| Veh2 | Tractor | E-W | |
| Veh3 | Farm truck | E-W | |
| Veh4 | Farm tractor w/trailer | W-E | |
| Veh5 | Tractor 3.5 tn | E-W | |
| Veh6 | Farm truck | E-W | Full load |
| Veh7 | Cargo truck 2.0 tn | E-W | |
| Veh8 | Farm truck | E-W | |
| Veh9 | Van | W-E | |
| Veh10 | Cargo truck 2.0 tn | W-E | |
| Veh11 | Van | W-E | Higher velocity |
| Veh12 | Passenger car | W-E | |
| Veh13 | SUV | W-E | Higher velocity |
| Veh14 | Passenger car | W-E | |
| Veh15 | Farm truck | E-W | |
| Veh16 | Farm truck | W-E | |
| Veh17 | Farm tractor w/trailer | E-W | |
| Veh18 | Passenger cars | W-E | 2 vehicles on the bridge |
| Veh19 | Farm truck | W-E | |
| Veh20 | Farm truck w/trailer | W-E | |
| Veh21 | Passenger car | E-W | |
| Veh22 | Farm truck | E-W | |
| Veh23 | Van | E-W | |
| Veh24 | Farm truck | W-E | |
| Veh25 | Fuel truck | E-W | Full load |
| Veh26 | Passenger car | E-W | Higher velocity |
| Veh27 | Passenger car | W-E | |

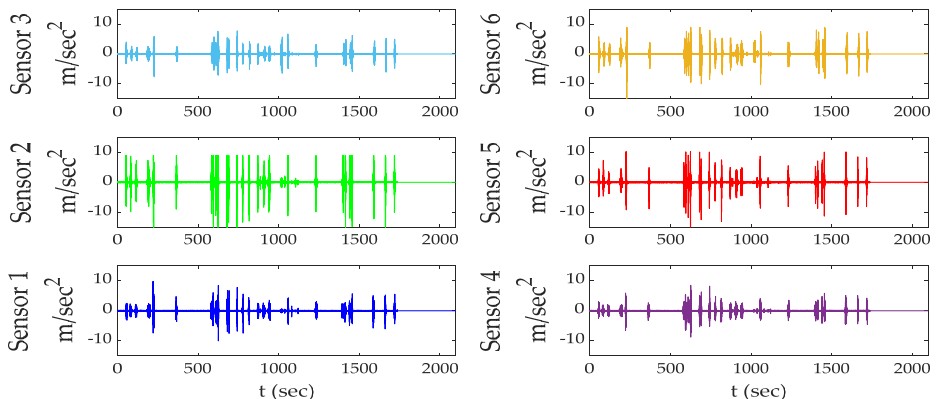

**Figure 9.** Recorded acceleration of total time histories in the vertical direction (axis Z) per sensor.

To facilitate the comparison, the time history plots displayed in Figure 9 refer to the sensors' arrangement of Figure 7. Examining and evaluating time histories is challenging due to their length. The fact that some vehicles cross the bridge close to each other affects the identification process of its modal parameters. For this reason, the first method of determining the first natural frequency was carried out using the free vibration time history segment of one vehicle case. Some researchers [31] use parts of time histories that they identified as ambient excitation, but this was not possible in this study due to the lack of environmental influence. Consequently, for sensors 2 and 5 (located in the middle of the bridge) time histories of vehicles Veh6 and Veh20 (Figure 10) were selected to be analyzed.

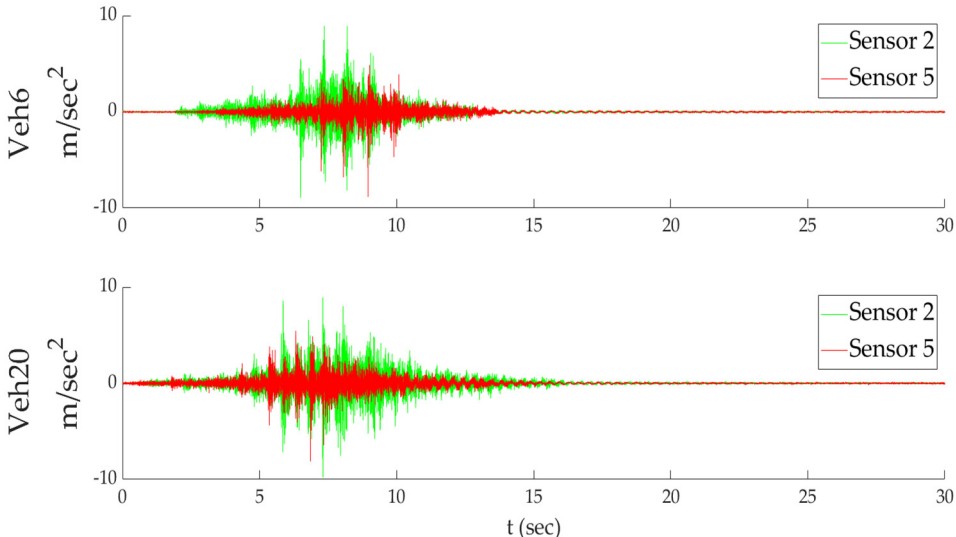

**Figure 10.** Partial time histories of sensors 2 and 5.

In Figure 10, the four partial time histories recorded at the middle of the bridge exhibit some differences. There are three possible reasons for these noisy time histories, namely, the effect of vehicle self-frequency combined with deck roughness, the issues related to bridge connections, and the eccentric passage of the vehicle with respect to the bridge's axis of symmetry. For this purpose, before extracting the segment related to the free vibration of these cases, a third-order Butterworth bandpass filter is assigned with cut-off frequencies at $f_{1c}$ = 2 Hz and $f_{2c}$ = 3 Hz, which comprises a frequency range estimation including the first natural frequency. In Figure 11, the partial filtered time histories are presented, while the vertical black line corresponds to the cut-off point of pure free vibration.

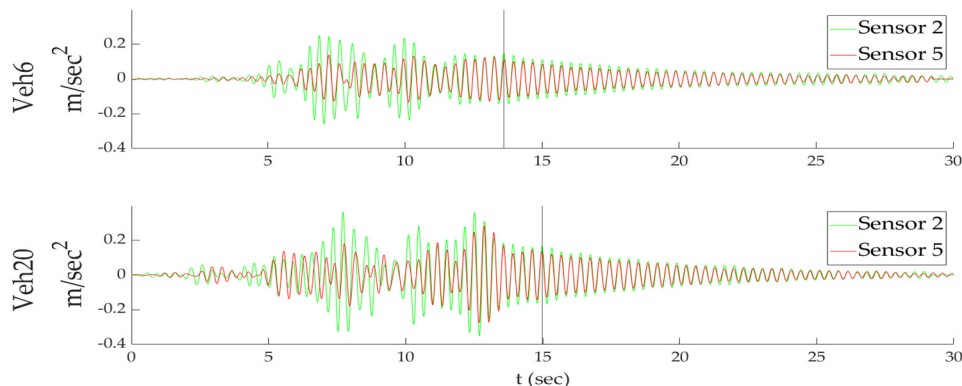

**Figure 11.** Partial filtered time histories of sensors 2 and 5.

As illustrated in Figure 12, in the case of vehicle Veh6, the natural frequency differs by 0.02 Hz between sensors 2 and 5. This analysis variance is insignificant compared to the used length of time history.

The damping ratio ($\zeta$) [50] is calculated from the equation:

$$\zeta = 1/(2\pi j) \cdot \ln\left(\ddot{u}_k / \ddot{u}_{k+1}\right) \tag{1}$$

The terms $\ddot{u}_k$ and $\ddot{u}_{k+1}$ indicate the acceleration values of the free vibration segments (Figure 11) that are separated by j repetition cycles. In Table 10, the results of six cases with j = 10 repetition cycles from sensors 2 and 5 are listed.

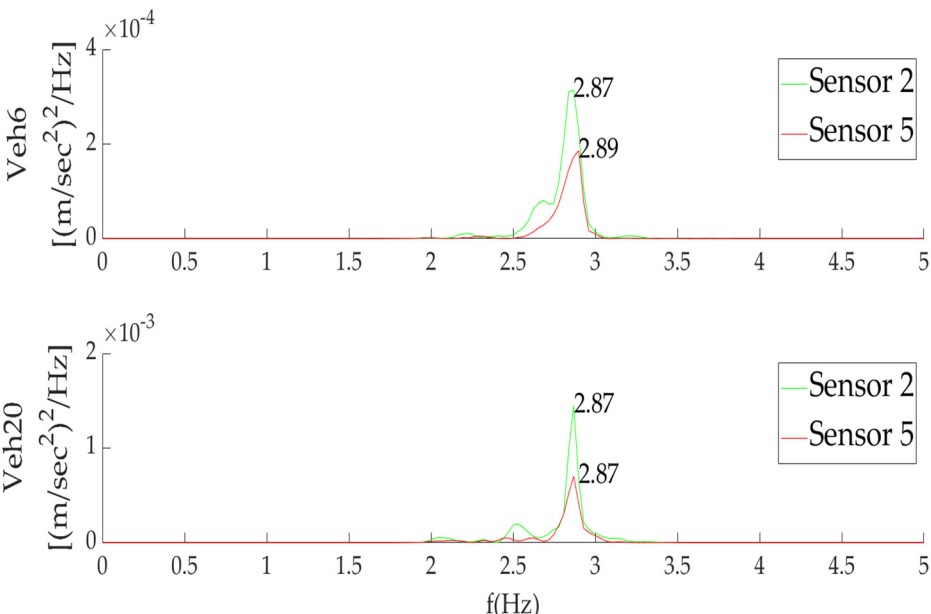

**Figure 12.** Single Side Power Spectral Density for the free vibration time histories of Veh6 and Veh20.

**Table 10.** Estimated damping ratios based on free vibration time histories.

| Vehicle | Sensor 2 | Sensor 5 |
|---|---|---|
| | $\zeta$ (%) | $\zeta$ (%) |
| Veh1 | 1.10 | 0.960 |
| Veh4 | 1.05 | 1.05 |
| Veh5 | 1.11 | 1.18 |
| Veh6 | 0.90 | 1.07 |
| Veh17 | 0.98 | 1.17 |
| Veh20 | 0.96 | 0.90 |
| Mean value | 1.02 | 1.05 |

As observed in Table 10, the damping ratios are almost equal to unity, meaning that the bridge vibration damps out slowly. For this reason, it is deduced that the close passage of vehicles affects the free vibration response of the bridge, consequently causing difficulty in determining the modal parameters of the bridge. The FDD method is then applied using the AFDD algorithm to determine the eigenfrequencies per array of sensors. The array of sensors 1-2-3 is used to determine the eigenfrequencies of the south girder (Side B, Figure 7), while the array of sensors 4-5-6 is used to determine the eigenfrequencies of the north girder (Side A, Figure 7). The independent process of examining the two girders originated from the assumption that connection conditions affect the girders' response independently.

Table 11 shows the identified frequencies for various loading cases. The "Veh" term's indexes refer to the group of vehicles passing the bridge and are related to the crossing distance of the vehicles over the bridge. For each loading case, the corresponding frequency (1st, 2nd... 6th) for both arrangements of the sensors presents variations up to 10%. These variations are attributed to differences in the structural integrity of the two girders' (mainly loose member connections). In addition, the variations presented for the same estimated frequencies between the loading cases are due to features of the crossing vehicles (bridge centerline passing, velocity). When the bridge passing takes place under normal conditions (low relative speed, one vehicle per time), then the mode shape refers to the 1st eigenmode, which is within the range of 2.79 to 2.89 Hz, with the value of 2.87 Hz being dominant. Additionally, it is found that the frequencies related to lateral excitation, in the range between 4 and 5 Hz, also appear. When many vehicles pass over the bridge, the bridge is excited in mode shapes found in all frequency ranges.

**Table 11.** Identified frequencies of the preprocessed raw measurements with FDD method in the vertical direction (Z axis).

| Case | AFDD at Sensors 1-2-3 Identified Frequencies (Hz) | | | | | | AFDD at Sensors 4-5-6 Identified Frequencies (Hz) | | | | | |
|---|---|---|---|---|---|---|---|---|---|---|---|---|
| | 1st | 2nd | 3rd | 4th | 5th | 6th | 1st | 2nd | 3rd | 4th | 5th | 6th |
| Veh1-2-3 | 2.87 | 3.36 | 5.98 | 6.67 | 9.27 | 13.3 | 2.87 | 3.49 | 5.76 | 6.70 | 8.71 | 11.3 |
| Veh4-5 | 2.82 | 3.40 | 4.39 | 6.93 | 8.51 | 11.8 | 2.76 | 3.49 | 4.36 | 5.11 | 8.51 | 12.1 |
| Veh6 | 2.83 | 6.52 | 7.23 | 8.84 | 9.85 | 11.5 | 2.86 | 7.69 | 9.20 | 9.98 | 11.1 | 12.0 |
| Veh7-8-9-10 | 2.83 | 4.54 | 6.26 | 9.12 | 10.3 | 13.8 | 2.81 | 4.36 | 7.65 | 9.08 | 10.5 | 14.6 |
| Veh11-12 | 2.86 | 3.91 | 5.59 | 9.08 | 12.0 | 14.2 | 2.86 | 6.12 | 8.54 | 10.2 | 11.3 | 16.2 |
| Veh13 | 2.83 | 3.40 | 4.39 | 6.93 | 8.51 | 11.8 | 2.76 | 3.49 | 4.36 | 5.11 | 8.51 | 12.1 |
| Veh14 | 2.87 | 5.50 | 7.59 | 8.53 | 9.32 | 11.1 | 2.87 | 5.35 | 6.39 | 8.51 | 10.4 | 11.9 |
| Veh15 | 2.86 | 4.83 | 6.25 | 8.59 | 9.23 | 11.3 | 2.83 | 3.97 | 6.18 | 8.60 | 10.1 | 11.4 |
| Veh16 | 2.88 | 6.25 | 8.41 | 9.61 | 11.0 | 13.8 | 2.88 | 3.43 | 7.05 | 8.41 | 10.8 | 11.7 |
| Veh17-18 | 2.79 | 3.38 | 4.87 | 6.22 | 9.46 | 11.0 | 2.79 | 3.44 | 4.52 | 4.88 | 8.44 | 11.2 |
| Veh19 | 2.88 | 4.91 | 6.42 | 8.33 | 12.5 | 13.5 | 2.89 | 4.16 | 5.26 | 8.33 | 9.95 | 12.6 |
| Veh20 | 2.86 | 5.46 | 6.61 | 7.59 | 9.45 | 11.1 | 2.86 | 4.99 | 7.01 | 8.79 | 10.5 | 12.8 |
| Veh21-22-23-24 | 3.37 | 5.85 | 8.70 | 10.1 | 11.2 | 14.0 | 2.70 | 3.43 | 4.62 | 7.06 | 8.69 | 12.1 |
| Veh25 | 3.11 | 4.58 | 5.65 | 7.52 | 8.74 | 9.42 | 3.11 | 3.42 | 5.37 | 8.76 | 10.1 | 11.6 |
| Veh26 | 2.81 | 4.21 | 8.40 | 9.10 | 11.5 | 14.5 | 2.83 | 4.49 | 6.55 | 8.70 | 10.6 | 14.6 |
| Veh27 | 2.50 | 4.47 | 7.08 | 9.26 | 10.2 | 13.9 | 2.89 | 5.80 | 7.42 | 8.52 | 11.3 | 12.8 |
| Veh1-27 | 2.83 | 4.52 | 6.22 | 7.57 | 8.52 | 9.41 | 2.78 | 3.46 | 4.53 | 4.89 | 7.05 | 8.52 |

In Table 12, the identified damping ratios show a wide range of values even for the same identified frequencies. This variance is due to various causes, such as the noisy recorded time histories and the spectrum calculation algorithm. Long-term time histories improve the resolution [51] and are more effective for signal analysis. Due to a lack of pure lateral excitation, determining eigenfrequencies in the transverse direction (Y-axis) by analyzing the free vibration segment of time histories is inefficient. Thus, by applying the FDD algorithm, the identified frequencies are shown in Table 13.

**Table 12.** Identified damping ratios with FDD method in the vertical direction (Z axis).

| Case | AFDD at Sensors 1-2-3 Identified ζ (%) | | | | | | AFDD at Sensors 4-5-6 Identified ζ (%) | | | | | |
|---|---|---|---|---|---|---|---|---|---|---|---|---|
| | 1st | 2nd | 3rd | 4th | 5th | 6th | 1st | 2nd | 3rd | 4th | 5th | 6th |
| Veh1-2-3 | 2.79 | 2.6 | 4.6 | 4.2 | 3.9 | 3.9 | 2.98 | 0.82 | 3.21 | 3.45 | 3.28 | 4.78 |
| Veh4-5 | 1.48 | 2.66 | 1.03 | 2.03 | 0.68 | 4.27 | 1.71 | 2.61 | 0.95 | 3.29 | 0.53 | 3.91 |
| Veh6 | 1.87 | 2.97 | 1.86 | 0.89 | 1.13 | 0.76 | 2.47 | 3.06 | 2.69 | 2.63 | 0.72 | 0.73 |
| Veh7-8-9-10 | 1.00 | 4.59 | 4.80 | 4.26 | 4.33 | 3.96 | 1.79 | 4.53 | 4.01 | 3.28 | 4.43 | 1.50 |
| Veh11-12 | 0.61 | 4.92 | 2.71 | 3.04 | 3.71 | 2.57 | 0.59 | 4.51 | 1.45 | 2.27 | 4.17 | 3.03 |
| Veh13 | 1.82 | 2.83 | 1.00 | 1.89 | 0.59 | 4.17 | 1.68 | 2.76 | 0.90 | 3.25 | 0.56 | 3.82 |
| Veh14 | 0.81 | 2.30 | 1.74 | 4.08 | 3.05 | 2.00 | 0.65 | 3.39 | 4.41 | 1.52 | 4.22 | 4.03 |
| Veh15 | 1.31 | 3.75 | 3.89 | 0.86 | 0.79 | 3.42 | 3.14 | 3.56 | 4.85 | 0.68 | 1.13 | 2.65 |
| Veh16 | 0.95 | 2.94 | 4.31 | 2.81 | 0.93 | 1.46 | 2.86 | 2.66 | 3.58 | 1.77 | 1.27 | 1.61 |
| Veh17-18 | 1.97 | 0.64 | 1.36 | 1.41 | 3.16 | 0.44 | 1.90 | 0.73 | 1.22 | 1.29 | 0.98 | 4.13 |
| Veh19 | 0.77 | 2.81 | 2.76 | 3.50 | 1.03 | 0.90 | 0.77 | 2.81 | 2.76 | 3.50 | 1.03 | 0.90 |
| Veh20 | 0.93 | 1.89 | 1.25 | 1.39 | 0.76 | 0.66 | 1.57 | 2.79 | 1.15 | 1.00 | 4.03 | 3.67 |
| Veh21-22-23-24 | 4.07 | 4.30 | 2.85 | 4.33 | 4.56 | 2.36 | 4.57 | 1.87 | 3.60 | 4.55 | 2.48 | 3.59 |
| Veh25 | 1.83 | 1.40 | 2.30 | 2.59 | 0.82 | 0.84 | 0.99 | 0.83 | 2.48 | 0.78 | 4.12 | 0.94 |
| Veh26 | 1.73 | 3.18 | 1.80 | 1.64 | 1.51 | 2.06 | 1.06 | 4.26 | 4.57 | 2.96 | 3.09 | 1.70 |
| Veh27 | 3.50 | 2.84 | 1.75 | 3.37 | 2.42 | 1.02 | 1.87 | 3.53 | 2.96 | 1.34 | 4.27 | 3.76 |
| Veh1-27 | 3.20 | 3.57 | 4.73 | 4.68 | 4.26 | 4.03 | 3.58 | 2.33 | 1.29 | 1.43 | 4.73 | 3.60 |

**Table 13.** Identified frequencies with FDD method in the transverse direction (Y axis).

| Case | AFDD at Sensors 1-2-3 Identified Frequencies (Hz) | | | | | | AFDD at Sensors 4-5-6 Identified Frequencies (Hz) | | | | | |
|---|---|---|---|---|---|---|---|---|---|---|---|---|
| | 1st | 2nd | 3rd | 4th | 5th | 6th | 1st | 2nd | 3rd | 4th | 5th | 6th |
| Veh1 … 27 | 3.47 | 4.90 | 6.25 | 8.93 | 10.7 | 11.9 | 3.39 | 4.91 | 6.71 | 7.47 | 8.53 | 11.7 |

Thus, determining the mode shapes of the determined frequencies is challenging. Figures 13–15 show the analysis of time histories 2 and 5 with the Short-Time Fourier Spectrum (STFT) method. The vertical yellow lines indicate the spectral distribution caused by vehicle excitations at the corresponding time.

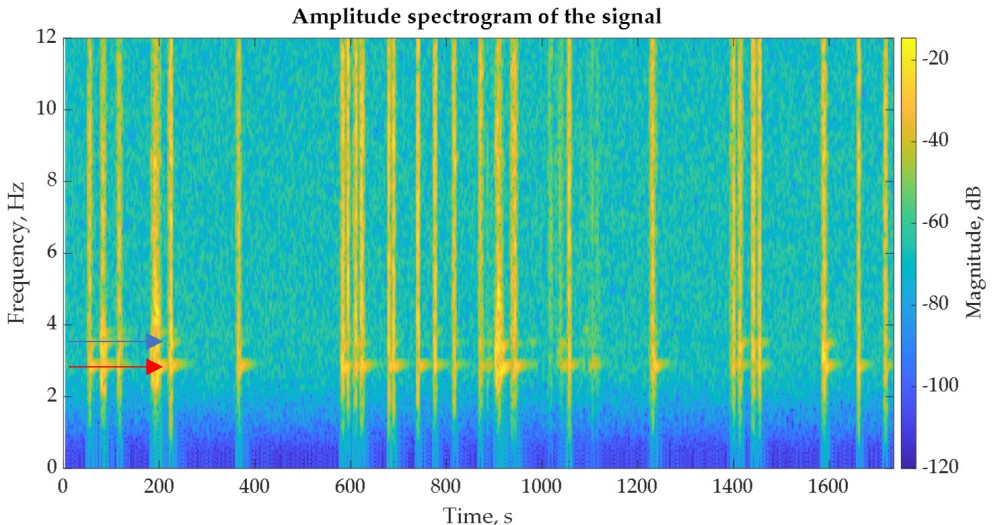

**Figure 13.** Short Time Fourier Spectrum of sensor 2 at vertical axis (Z axis).

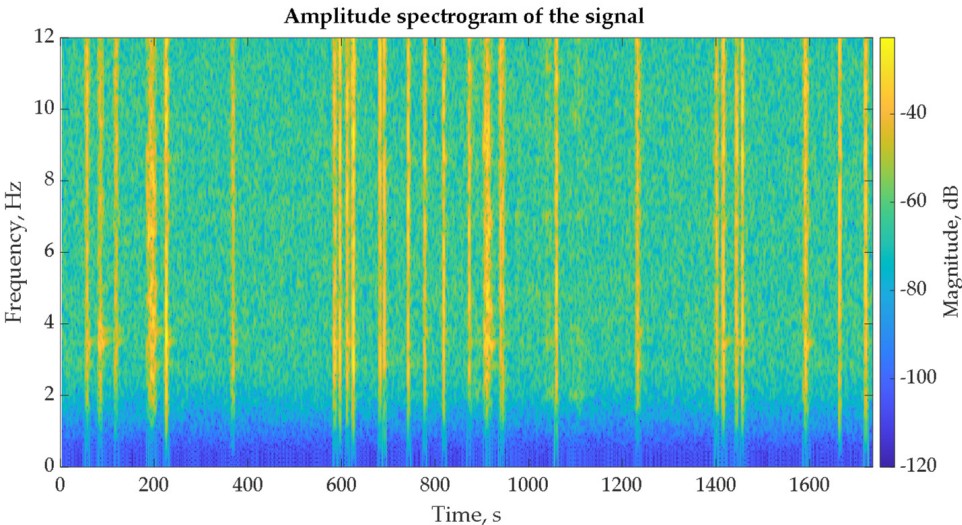

**Figure 14.** Short Time Fourier Spectrum of sensor 2 at transverse axis (Y axis).

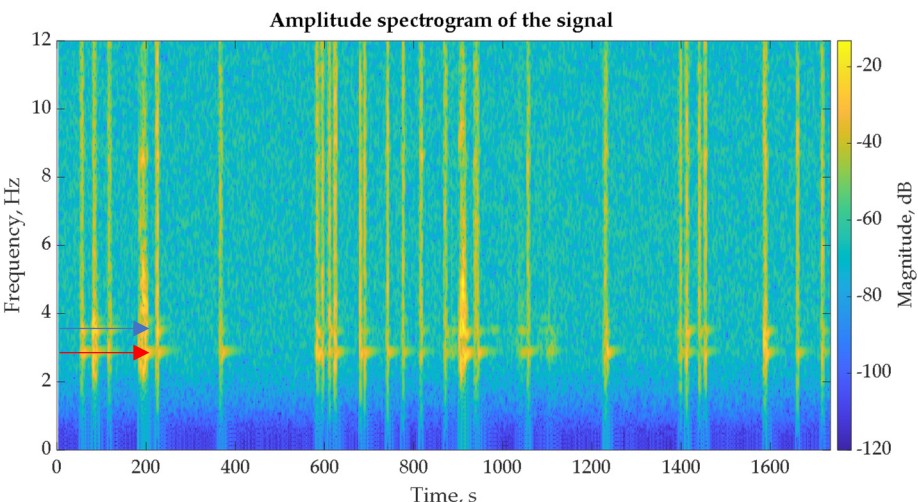

**Figure 15.** Short Time Fourier Spectrum of sensor 5 at vertical axis.

Figures 13 and 16 show that the frequencies at 2.87 and at 3.42 Hz appear on the vertical axis for both sensors (red and blue arrows, respectively, in Figures 13 and 15). The frequency at 2.87 Hz appears in all time histories, while the frequency at 3.42 Hz appears in the case of vehicles entering the bridge at close distances. Both frequencies appear on the vertical axis, while the frequency at 3.42 Hz also appears on the transverse axis, demonstrating 1st and 2nd mode shapes identified as bending and torsional bending, respectively. The other eigenfrequencies that correspond to additional mode shapes are not easily distinguished.

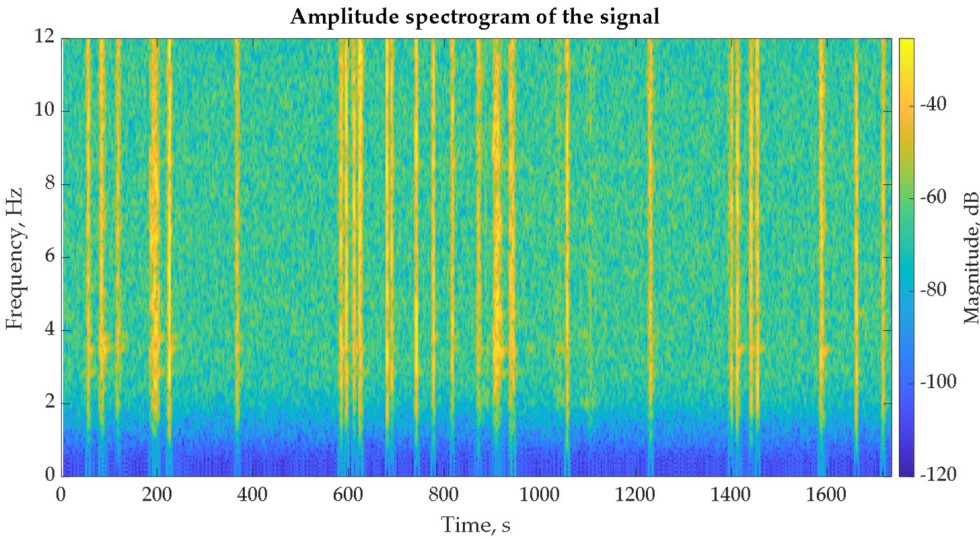

**Figure 16.** Short Time Fourier Spectrum of sensor 5 at transverse axis (Y axis).

Based on the identified frequencies, the graphical display of the mode shapes is a preliminary check for their reliable assessment. The procedure for the static graphical display of the mode shapes based on the determined frequencies is described in detail in the literature [49]. For this reason, a set of frequencies is selected, which is assumed to correspond to distinct mode shapes. The two clearly identified frequencies at 2.87 and 3.42 Hz are used for the graphical display, as determined by the FDD and STFT methods.

The remaining frequencies are selected from a range of frequencies as specified in Table 11 by means of an iterative process. Estimating the frequencies corresponding to real mode shapes is extremely difficult due to their scattering, as evidenced in Table 11. The chosen frequencies are 2.87, 3.42, 4.57, 7.81, and 8.54 Hz, and one of the sensors is selected as

the reference channel. The next step is calculating the auto-power spectrum of the reference sensor and the cross-power spectra between the pairs of reference-remaining sensors. The amplitude and phase are calculated for each case, and then normalization is carried out, so that the mode shape with the largest amplitude has a unity value. With sensor 4 as the reference, the shapes of the group of frequencies are shown in Figures 17–21, respectively. It is also noticed that the shapes are not clearly distinguished due to several reasons that are outside of the scope of this study. It is finally established that the determined frequencies with the FDD and STFT methods yield the dynamic response of the bridge, as evidenced by the graphically represented mode shapes.

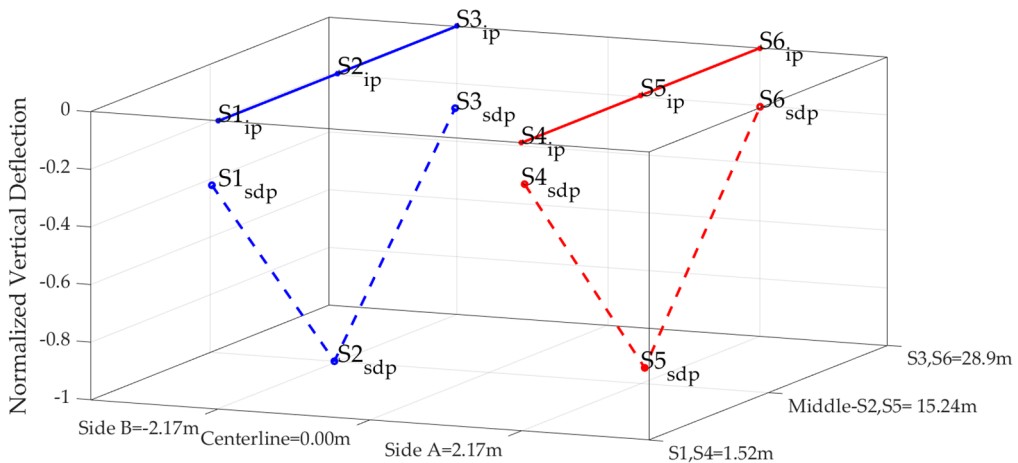

**Figure 17.** 1st Identified mode shape—bending with sensor 4 as reference at 2.87 Hz.

where ip: node initial position and sdp: node static display position.

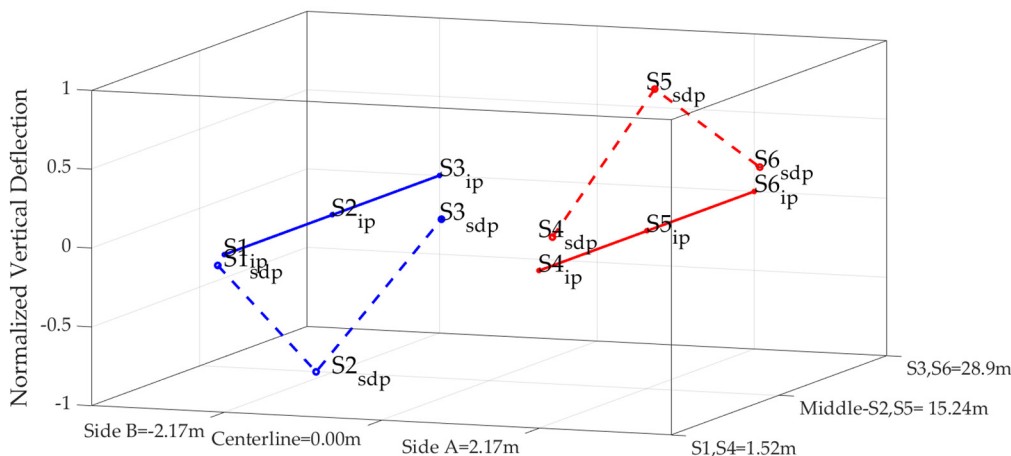

**Figure 18.** 2nd Identified mode shape—torsional with sensor 4 as reference at 3.42 Hz.

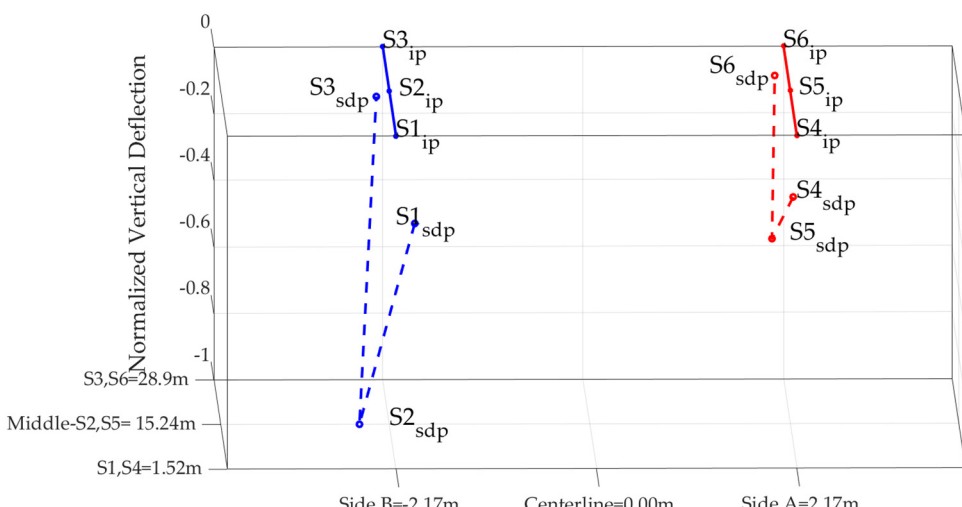

**Figure 19.** 3rd Identified mode shape—lateral with sensor 4 as reference at 4.57 Hz.

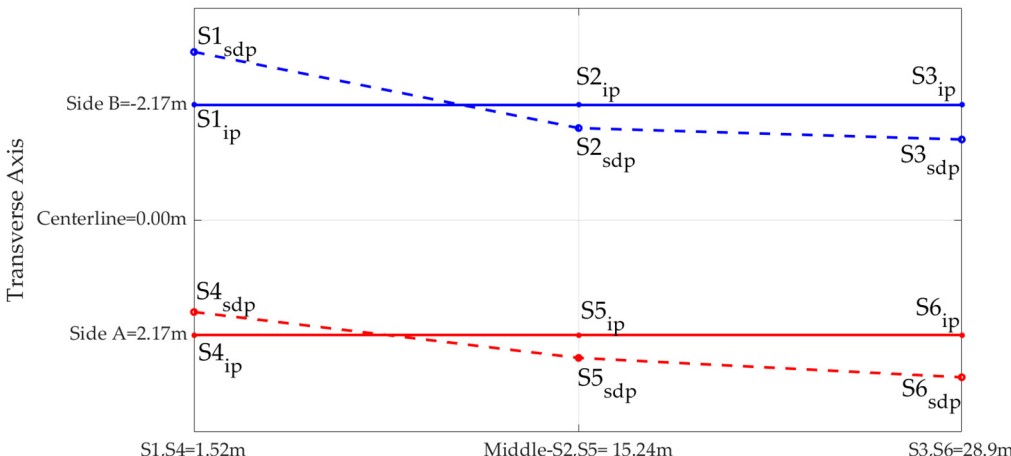

**Figure 20.** 4th Identified mode shape-2nd lateral with sensor 4 as reference at 7.81 Hz.

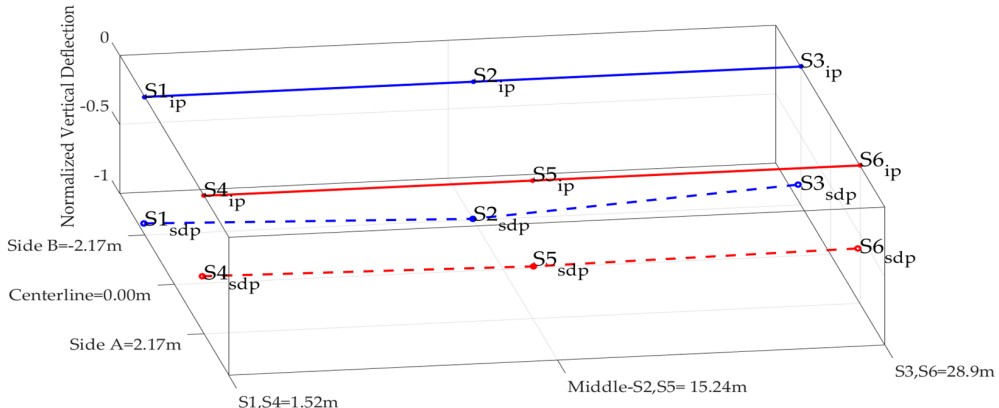

**Figure 21.** 5th Identified mode shape-2nd bending with sensor 4 as reference at 8.54 Hz.

Figures 17–21 show the static display of the bridge at the selected frequencies, which are estimated to correspond to eigenmodes. However, the sensor positions show minor deviations for each case. For example, for the 1st eigenmode, the normalized position of sensor 5 is at $-0.95$, whereas it should be at $-1$, as sensor 2. This difference is due to the algorithm calculating the reference sensor autocorrelation and the remaining sensors' cross-correlation. The number of time-history points used to calculate the autocorrelation

and cross-correlation affects the resolution of the results. Moreover, the "noise" of the time histories and their pre-processing also affect the amplitude of the auto-power spectral density and the cross-spectral density, which are used to normalize the position of the sensors. These variations can be considered as acceptable, since in Figures 17–21 the estimated eigenmodes are approximated very accurately, despite their present deviations. Although many factors affect the identification process of Figures 17–21, the modes are approached with good accuracy and fully agree with the results of the numerical analyses. A more meticulous process in recording the time histories, with dense instrumentation of the bridge in the longitudinal and vertical directions, is expected to lead to more accurate results.

### 3.2. Finite Element Modeling and Numerical Modal Analysis Results

In addition, a numerical finite element (FE) model was developed using the commercial CSIBridge software [52]. The parts' cross-sections of this bridge were determined according to the literature and on-site measurements. The main bridge parts were modeled by straight beam (frame) elements with six degrees of freedom at all nodes, as shown in Figure 22a.

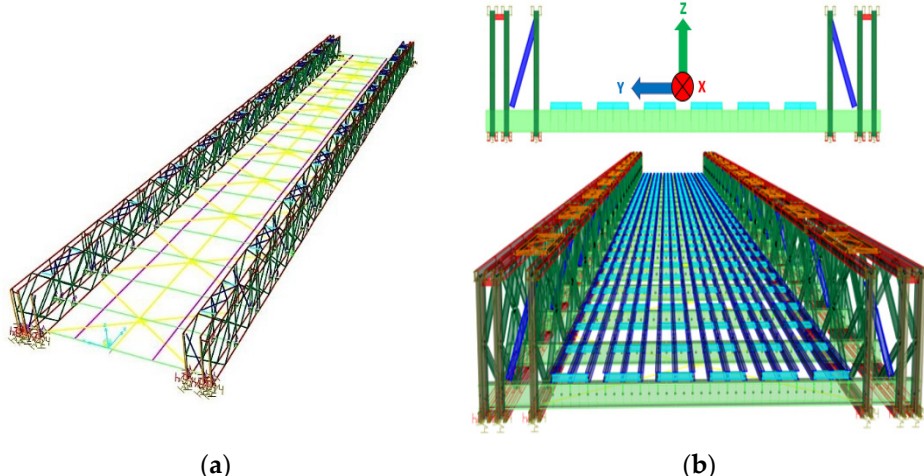

(**a**)                                                    (**b**)

**Figure 22.** TS Bailey bridge FE model: (**a**) straight frame model; (**b**) 3D visualization.

Horizontal and vertical links, which are constrained along the three axes, are used to connect the cross-beams (Figure 23) with the panels. The horizontal bracings were also modeled by links, with a sufficiently large stiffness taken equal to 25 MN/m.

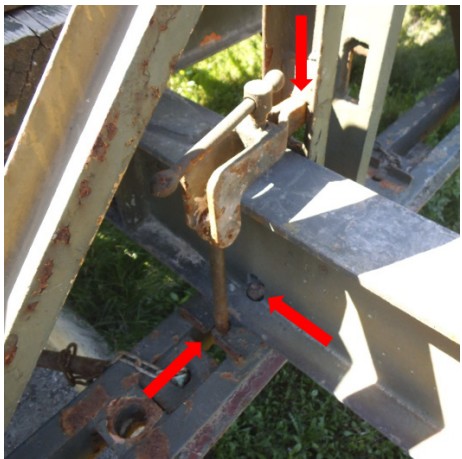

**Figure 23.** Panel—cross-beam connection.

The bridge's supports were simulated with twelve 6-degree-of-freedom springs (one for each bearing point). High stiffness values were assigned for the transversal springs of both bearing sides to restrain the corresponding degrees of freedom, while low stiffness values were assigned for the rotational ones to simulate the hinge joint.

Longitudinally, the bridge displacement at both sides is restrained due to developing friction forces between the bearing with the steel plate and the end vertical member (Figure 24). These restraining forces are simulated with springs (link elements).

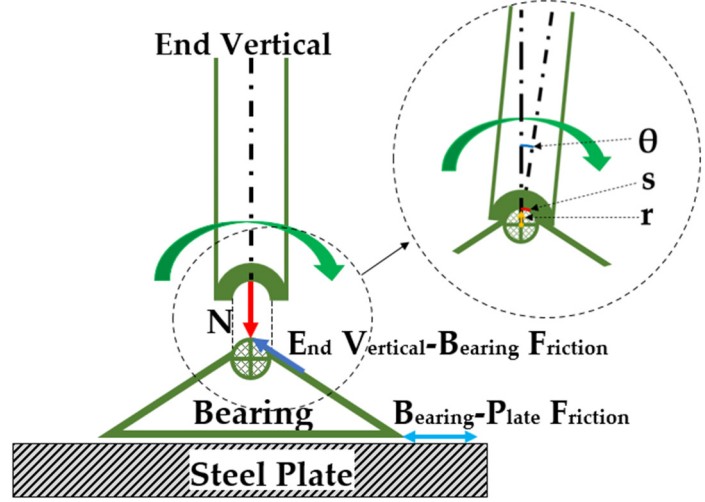

**Figure 24.** Bearing contact forces.

If the End Vertical—Bearing (EVB) friction is equal to the hypothetical corresponding spring forces, then the spring stiffness Kx is calculated according to the Equations (2)–(6):

$$EVB_{friction} = F_{spring} \tag{2}$$

$$EVB_{friction} = \mu \cdot N \tag{3}$$

where N is the vertical reaction at the bearing support, and $\mu$ is the friction coefficient, estimated between 0.42 and 0.50.

$$F_{spring} = Kx \cdot s \tag{4}$$

The term s in the above equation is calculated from the equation:

$$S = r \cdot \theta \tag{5}$$

where the term r is the radius of rotation and $\theta$ is the angle (Figure 24) corresponding to a simply supported beam and can be expressed by the equation:

$$\Theta = w \cdot L^3 / (24 E \cdot I) \tag{6}$$

where w is the uniformly distributed load, L is the bridge length, E is the elastic modulus, and I is the moment of inertia.

Substituting Equations (2), (3), (5), and (6) into Equation (4):

$$Kx = 24 \cdot \mu \cdot N \cdot E \cdot I / r \cdot w \cdot L^3 \tag{7}$$

The mass of the numerical model differs from the actual bridge mass because some secondary bridge parts were not simulated, without affecting the total bridge stiffness and dynamic response. The remaining mass was proportionally added at each node, while this proportionality was derived from each node's contribution to the total mass. Figure 25 shows the first six mode shapes derived from numerical model analysis.

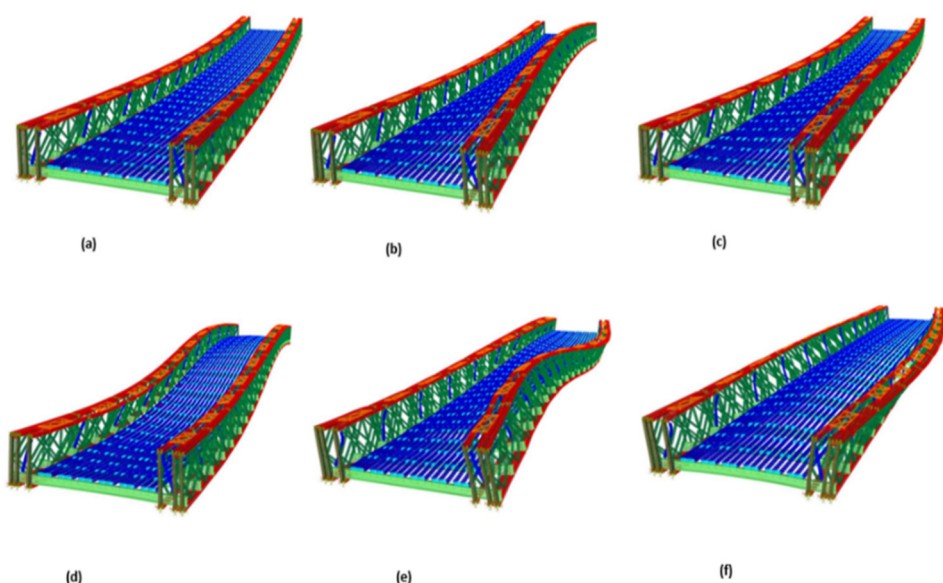

**Figure 25.** The first 6 eigenmodes of the FE model of the bridge (**a**) 1st: vertical bending at 2.87 Hz (**b**) 2nd: torsional-lateral at 3.22 Hz (**c**) 3rd: lateral-torsional at 4.38 Hz (**d**) 4th: second lateral at 7.50 Hz (**e**) 5th: second vertical bending at 8.73 Hz (**f**) 6th: non-identified at 9.87 Hz.

The direct-integration time-history dynamic analysis (linear case) was performed by simulating vehicles (modified vehicle type of CSIBridge) corresponding to those that crossed the actual bridge. The discretization time step was set to 0.02 sec ($f_{sampling}$ = 50 Hz), aiming at reducing the computational time by following the program guide to use a time step at least equal to 1/10 of the longest period, and considering that the expected frequencies are estimated in the $f_{Nyquist}$ range.

$$f_{Nyquist} = f_{sampling}/2 = 25\ Hz$$

The damping coefficient was proportionally set at 1% for all mode shapes based on the values in Table 10. The Table 12 values were considered to be overestimated, so they were not accounted for in the analysis.

Tables 14 and 15 show the frequencies identified with the FDD method in the vertical and transverse direction, respectively, as the results obtained from dynamic analysis of passage of the 27 vehicles. In the case of the centerline offset by 30 cm, the relevant results are presented in Tables 16 and 17.

**Table 14.** Identified frequencies of numerical results with FDD method in the vertical direction (Z axis).

| Case | AFDD at Sensors 1-2-3 Identified Frequencies (Hz) | | | | | | AFDD at Sensors 4-5-6 Identified Frequencies (Hz) | | | | | |
|---|---|---|---|---|---|---|---|---|---|---|---|---|
| | 1st | 2nd | 3rd | 4th | 5th | 6th | 1st | 2nd | 3rd | 4th | 5th | 6th |
| Veh1-27 | 2.85 | 3.47 | 7.98 | 8.83 | 10.3 | 11.9 | 2.85 | 3.44 | 7.98 | 8.83 | 10.3 | 11.9 |

**Table 15.** Identified frequencies of numerical results with FDD method in the transverse direction (Y axis).

| Case | AFDD at Sensors 1-2-3 Identified Frequencies (Hz) | | | | | | AFDD at Sensors 4-5-6 Identified Frequencies (Hz) | | | | | |
|---|---|---|---|---|---|---|---|---|---|---|---|---|
| | 1st | 2nd | 3rd | 4th | 5th | 6th | 1st | 2nd | 3rd | 4th | 5th | 6th |
| Veh1-27 | 3.17 | 4.29 | 7.99 | 9.48 | 10.3 | 17.4 | 3.18 | 4.29 | 6.99 | 7.99 | 9.48 | 10.3 |

**Table 16.** Identified frequencies of numerical results with FDD method in the vertical direction (Offset Lane).

| Case | AFDD at Sensors 1-2-3 Identified Frequencies (Hz) | | | | | | AFDD at Sensors 4-5-6 Identified Frequencies (Hz) | | | | | |
|---|---|---|---|---|---|---|---|---|---|---|---|---|
| | 1st | 2nd | 3rd | 4th | 5th | 6th | 1st | 2nd | 3rd | 4th | 5th | 6th |
| Veh1-27 | 2.85 | 3.16 | 4.28 | 7.99 | 8.83 | 10.3 | 2.85 | 3.19 | 4.29 | 7.98 | 8.83 | 10.3 |

**Table 17.** Identified frequencies of numerical results with FDD method in the transverse direction (Offset Lane).

| Case | AFDD at Sensors 1-2-3 Identified Frequencies (Hz) | | | | | | AFDD at Sensors 4-5-6 Identified Frequencies (Hz) | | | | | |
|---|---|---|---|---|---|---|---|---|---|---|---|---|
| | 1st | 2nd | 3rd | 4th | 5th | 6th | 1st | 2nd | 3rd | 4th | 5th | 6th |
| Veh1-27 | 3.18 | 4.28 | 7.01 | 7.98 | 9.48 | 10.3 | 3.18 | 4.28 | 7.01 | 7.98 | 9.48 | 10.3 |

## 4. Discussion

Despite the observed dispersion of the identified frequencies, the range of eigenfrequencies can be determined for the respective modes of operation with satisfactory accuracy. One of the issues that often causes such discrepancies is that, in OMA, the spectral distribution of both the bridge itself and the vehicles is recorded and represented in the analysis [19]. Thus, the mode shapes and the mode deflection shapes are obtained simultaneously. The fuzzy boundaries that separate them can be smoothed out by appropriate adjustments [49].

Another issue that affects the variations in the determined frequencies with the FDD method is the quality of the time histories, known as the signal-to-noise ratio (SNR). Noisy signals require processing to make them appropriate for further analysis. However, the noise source is due to either the equipment being used or the operational condition of the bridge. The operational condition is related to the bridge's structural health, part of which is the condition of the connections of its various parts. Inspection revealed loose connections on the deck and the B-side girder.

The differences in the spectral content of their time histories are compatible with this observed looseness, as shown in Figures 26 and 27, showing the spectral content of the two sensors for the case of one vehicle passing (Veh6) and adjustment of the horizontal axis up to the value of 16 Hz for the presentation needs.

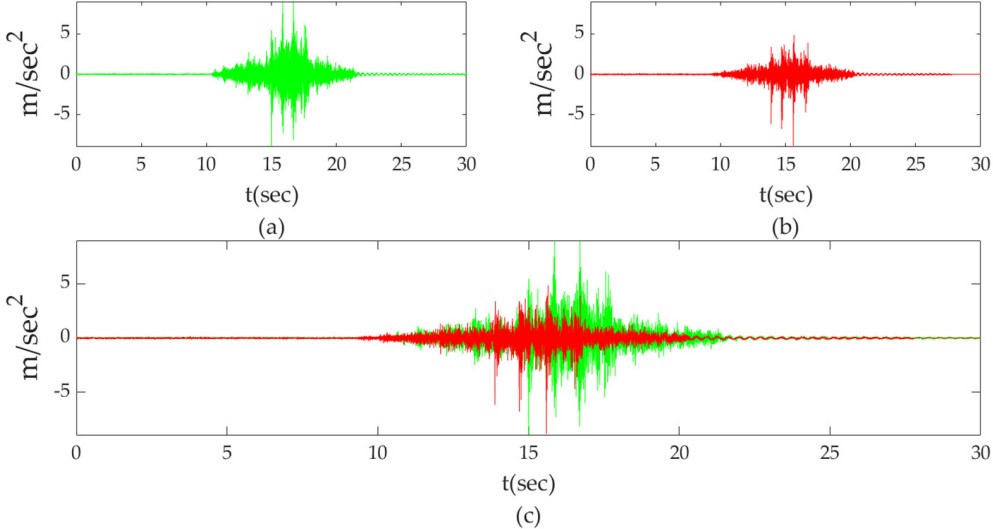

**Figure 26.** Measurement's comparison of sensors 2 and 5 for Veh6; (**a**) sensor 2; (**b**) sensor 5; (**c**) sensor 2 vs. sensor 5.

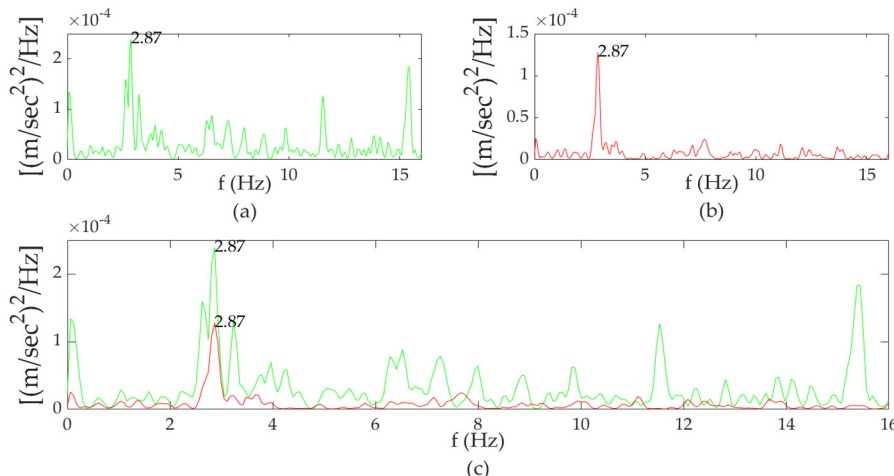

**Figure 27.** Single-sided Power Spectral Density of sensors 2 and 5: (**a**) sensor 2; (**b**) sensor 5; (**c**) sensor 2 vs. sensor 5.

Sensor 2 shows frequencies also found in sensor 5, but which are less noticeable, so it is concluded that the connections on side B differ compared to side A. Comparing the analysis results of the raw measurements with those of the numerical analysis, it is deduced that the numerical simulation satisfactorily approximates the dynamic response of the bridge. The identified mode shapes and the corresponding eigenfrequencies, which resulted from the numerical analysis, were found to approximate the mode shapes determined from the analysis of the raw measurements. However, the assumptions related to the loading of the vehicles and the velocities passing over the bridge certainly also affect the response of the bridge. To determine the unique characteristics of the bridge, it is necessary to carry out a controlled OMA, which helps to fully validate the numerical simulation.

An indicator for the validation of the FE numerical model is the similitude between the time histories recorded in the field and those resulting from the numerical analysis. Indicatively, Figure 28 shows the comparison for the case of Veh6. From the diagrams, it can be seen that all values of time histories are found in the same range, and the numerical results match the time histories of sensor 5 in spectral content. Sensor 2 exhibits frequencies in the entire range. This assumption confirms the abovementioned conclusion that the girder where sensor 5 is installed is in better structural condition than the one with sensor 2 installed.

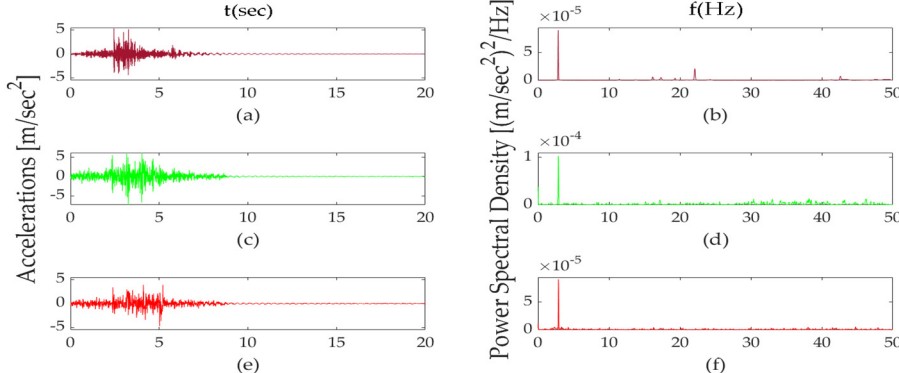

**Figure 28.** Comparison of the acceleration time histories and Power Spectral Density for Veh6 case: (**a**,**b**) numerical results at the middle of the bridge; (**c**,**d**) sensor 2; (**e**,**f**) sensor 5.

It is found that the determining frequencies for each case are almost the same with slight differences, as presented also in Table 18. The frequencies of the eigenmodes of real bridges are found in a frequency range for each mode shape. This finding may be attributed

to bridges exhibiting loose connections, which can change the spectral distribution for the same mode shapes each time. A re-evaluation of the bridge after required maintenance is vital to re-validate the FE model.

**Table 18.** Comparison of identified frequency results.

| Case Analysis | Identified Frequencies (Hz) | | | | | | Remarks | | | | | |
|---|---|---|---|---|---|---|---|---|---|---|---|---|
| | **1st** | **2nd** | **3rd** | **4th** | **5th** | **6th** | **1st** | **2nd** | **3rd** | **4th** | **5th** | **6th** |
| Raw measurements | 2.87 | 3.42 | 4.57 | 7.81 | 8.54 | Non-identified | FDD analysis | | | | | |
| FE modal analysis | 2.87 | 3.22 | 4.38 | 7.50 | 8.73 | 9.87 | Eigen frequencies | | | | | |
| Numerical analysis | 2.85 | 3.47 | 4.29 | 7.98 | 8.83 | 9.48 | Vertical and transverse axes | | | | | |

## 5. Conclusions

The dynamic response of a 30.48 m Triple-Single Bailey bridge was recorded using low-cost sensors under normal operating conditions. The recorded time histories of the bridge response were due to the passage of 27 vehicles. The OMA method and the AFDD algorithm determined five eigenfrequencies and their mode shapes. In addition, a FE model was developed, and numerical analyses were performed to assess the dynamic response of the bridge and determine the range of identified response frequencies.

From the results, it was found that the dynamic response determined by the numerical analysis satisfactorily corresponds to the real bridge response and can be used for further dynamic analyses. Deviations between the frequencies are justified due to noise from the bridge–vehicle interaction or the sensors' errors. Finally, it is accepted that a denser network of sensors would yield more reliable results.

**Author Contributions:** Conceptualization, C.J.G. and V.D.P.; methodology, C.J.G. and V.D.P.; software, V.D.P.; validation, V.D.P. and X.A.L.; formal analysis, C.J.G. and V.D.P.; investigation, C.J.G.,V.D.P. and P.T.; resources, C.J.G. and V.D.P.; data curation, V.D.P. and X.A.L.; writing—original draft preparation, C.J.G. and V.D.P.; writing—review and editing, P.T. and X.A.L.; visualization, V.D.P.; supervision, C.J.G.; project administration, C.J.G.; All authors have read and agreed to the published version of the manuscript.

**Funding:** This research received no external funding.

**Institutional Review Board Statement:** Not applicable.

**Informed Consent Statement:** Not applicable.

**Data Availability Statement:** Data are contained within the article.

**Conflicts of Interest:** The authors declare no conflict of interest.

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
