# Peer review of "Dynamic Response Identification of a Triple-Single Bailey Bridge Based on Vehicle Traffic-Induced Vibration Analysis"

_infrastructures, doi:10.3390/infrastructures7100139_

Round 1
Reviewer 1 Report
see attaches

Author Response
see attaches

Reviewer 2 Report
This paper presented a very practical study of Bailey bridges which were commonly used in emergency rescue. The results and conclusions are useful and instructive significance. Before the recommendation of this paper, revisions should be made considering the following comments.
(1) Full definition should be given for the abbreviation, such as 3D, AFDD et al.
(2) Parenthesis is missed in Line 73 for EMA.
(3) The citation formats should be corrected according to the journal guidance.
(4) Line 335, please delete the unit Hz by only remaining the last one.
(5) Lines 382-393, please correct the formats of equations and the relevant wordings.
(6) Please clearly define what is the operational conditions discussed in this study.
(7) Literature review is inadequate, following reference but not limit to this should be duly reviewed.
Recent Advances in Researches on Vehicle Scanning Method for Bridges doi.org/10.1142/S0219455422300051
Author Response
see attaches
